# Kinetic Electromagnetic Energy Harvester for Railway Applications—Development and Test with Wireless Sensor

**DOI:** 10.3390/s22030905

**Published:** 2022-01-25

**Authors:** Zdenek Hadas, Ondrej Rubes, Filip Ksica, Jan Chalupa

**Affiliations:** Faculty of Mechanical Engineering, Brno University of Technology, 616 69 Brno, Czech Republic; Ondrej.Rubes@vut.cz (O.R.); Filip.Ksica@vutbr.cz (F.K.); chalupa@fme.vutbr.cz (J.C.)

**Keywords:** energy harvesting, train, electromagnetic transducer, model, vibration, test, wireless sensor

## Abstract

This paper deals with a development and lab testing of energy harvesting technology for autonomous sensing in railway applications. Moving trains are subjected to high levels of vibrations and rail deformations that could be converted via energy harvesting into useful electricity. Modern maintenance solutions of a rail trackside typically consist of a large number of integrated sensing systems, which greatly benefit from autonomous source of energy. Although the amount of energy provided by conventional energy harvesting devices is usually only around several milliwatts, it is sufficient as a source of electrical power for low power sensing devices. The main aim of this paper is to design and test a kinetic electromagnetic energy harvesting system that could use energy from a passing train to deliver sufficient electrical power for sensing nodes. Measured mechanical vibrations of regional and express trains were used in laboratory testing of the developed energy harvesting device with an integrated resistive load and wireless transmission system, and based on these tests the proposed technology shows a high potential for railway applications.

## 1. Introduction

Modern railways are required to provide an improved quality of service and high levels of safety. Reliable trackside infrastructure maintained in good condition is important for smooth transportation of goods and passengers. To accomplish that, preventive maintenance and scheduled maintenance techniques are currently being used for trackside infrastructure, which can reveal critical wears, defects or failures. However, continuous condition monitoring and long-time sensing using modern electronics could detect incipient wears, failures and degradation that could affect safe railway operation.

Monitoring of trackside systems is important in order to reveal significant changes in functional parameters (e.g., deformation, vibration and temperature). This type of monitoring and diagnostics is widely known as condition-based maintenance, and its main goal is to provide significant savings in infrastructure operational costs. Predictive maintenance techniques require detailed trackside monitoring and the employment of many sensing systems. Reliable and low-maintenance power supplies are essential prerequisites to reliable predictive maintenance results.

Electrical power for these monitoring systems could be delivered from a catenary that is a part of the trackside infrastructure electrical grid. The catenary, however, could be difficult to access due to tight restrictions set up by the infrastructure management and operation in order to maintain the reliability of the track systems. Furthermore, even in the case of a modern railway network, many electrical systems access points are still remote or quite difficult to access due to poor infrastructure and a lack of foresight in regard to modern wireless sensing system power management. Cables and wires are an expensive part of the infrastructure, are often subject to theft, and are difficult to maintain, especially when the layout of railway tracks is changed. Auxiliary railway systems are ready to accept alternative power sources and achieve economically efficient operation by using alternative and energy harvesting sources to power them. Renewable energy sources, such as solar panels or wind turbines, could be used for remote applications that are quite demanding in terms of their power consumption (e.g., warning and signal lights, track switches, grade crossing signals, point machines, positive train control systems and train positions, communication access points etc.). Other energy harvesting sources are widely discussed for embedded monitoring systems in railways. Energy harvesting has been used for wireless sensor nodes and low-power autonomous systems for over 20 years [1]. In general, energy harvesting is based on the conversion of ambient energy into useful electricity. In trackside environment, the passing train by itself could deliver a wide variety of ambient mechanical energy (e.g., mechanical vibration, rail deformation, the sag of sleepers or rails etc.) that could be utilized for such systems.

Individual trackside energy harvesting technologies are summarized in this paper, and the physical principle of a kinetic energy harvesting solution based on converting track vibration into electricity is proposed. On the basis of a mathematical model, a design for a maintenance-free kinetic energy harvester is developed and described, including experimental results and the testing of a complete system with a sensor node.

## 2. Energy Harvesting Technologies for Trackside Applications

Recent developments in wireless technologies have resulted in a significantly smaller size, lower price and decreased energy consumption of these systems. For this reason, in railways applications wired sensors are often being abandoned and replaced by wireless alternatives. Their main advantages, on top of the abovementioned ones, are their easy installation and simplified maintenance. The primary battery source and operation in low-power mode could assure the reliable operation of these sensors for more than a year [2]. However, such a period is still close to the required maintenance period of the sensor itself. For this reason, energy harvesting technologies are investigated in order to achieve several years of maintenance-free operation of these sensor nodes.

Wireless sensor nodes with autonomous energy harvesters could find their way into various engineering applications, as they could operate autonomously in maintenance-free mode for long periods of time. As an example, these solutions are currently used in heavy industry applications, structural health monitoring systems [3], aerospace [4] and transportation [5]. Current energy harvesting technologies have been investigated as a possible source of power for wireless applications, which would otherwise be difficult to connect to the existing power grid.

Track condition monitoring applications are developed on the basis of acceleration sensors [6] or strain gauges [7], mainly for the condition monitoring of a crossing [8,9,10]. Bridge monitoring solutions have also been widely discussed in recent publications. Paper [11] discussed the feasibility of a bridge monitoring system in terms of its operational life. It illustrated how the traffic on a bridge over time could accentuate the identification of damage, which was necessary to know the state and health of the structure. A segmental prefabrication and assembly of the bridge on the Guangzhou Metro was presented in [12]. Passing vehicles induced vibrations used for energy harvesting, and using appropriate modelling and dynamic analyses of the bridge system a new type of electromagnetic vibration energy harvester was proposed. This device was designed in a way that could power strain-collection units for a bridge health monitoring system. These typical sensing applications, such as crossings and bridges, provide a measurable dynamic response of the track infrastructure to the passing train. In this case, the measured response from the passing train could serve as a suitable source of energy for monitoring applications.

Many published papers dealing with trackside energy harvesting solutions showed that harvesting energy from passing trains has a great potential for wireless sensing applications in railways. Authors of paper [13] investigated the possibility of establishing a self-powered wireless sensor network by integrating the ZigBee stack protocol together with an energy harvesting power source. This system is used for the condition monitoring of urban rail transit utilizing localized energy harvesting. Authors of the previous article also present another complex system for the smart monitoring of an underground railway by local energy generation in paper [14]. 

A study and the results of a portable electromagnetic energy harvesting system are presented in papers [15,16]. Their proposed solution consists of a mechanical transmission and an electrical regulator that converts sags in the rail into electricity, providing a peak voltage of 58 V at 1 Hz with a displacement of 2.5 mm. Authors from Stony Brook presented a preliminary prototype of a mechanical-rectifier-based harvester [17]. This study illustrated that sufficient power can be harvested by the device, which is based on a motion rectifier design. A novel direct-motion-driven harvester was described in publication [18], where the authors describe how an anchorless mounting results in a higher power capacity without the requirement of any special preparation during its installation. An installation and test under a fully loaded freight train running at 64 km/h was also presented in this paper. Paper [19] presented a design, modelling, in-lab experiment and field-test results of a mechanical motion rectifier mechanism which is based on a compact ball-screw-based electromagnetic energy harvester. A theoretical study of a cam mechanism was presented by the University of Nebraska in publication [20], where it was used to exploit the contact between a train wheel and a harvester mechanism to drive an electromagnetic generator. A solution based on a direct load piezoelectric harvesting device was proposed in paper [21], offering a structurally simple solution in the form of a piezoelectric drum device placed under sleepers. Piezoelectric solutions for strain-based energy harvesting have been widely discussed, where piezoceramic patches or piezo stacks transduce deformation into electricity [22]. 

The abovementioned technology mainly converts a direct train load in form of direct contact, deformation or element strain. These devices have the potential to provide peak output power of several watts; however, they are not suitable for high-speed rail applications due to their necessity for a mechanical contact. In contrast, the subsequently presented drum and patch type piezoelectric element solutions are suitable for high-speed operation at the cost of a lower power output in the range of several microwatts. These piezoelectric elements provide a very high voltage but a low current. This disadvantage could be eliminated by multilayer piezoelectric composites; however, the manufacturing of such materials is a very expensive process and for this reason it is not suitable for cost-effective wireless sensor nodes.

Kinetic energy harvesting solutions capable of transducing kinetic energy from vibrations under the passing train into electricity serve as maintenance-free sources of energy. Authors of publication [23] investigated the possibility of harvesting energy from the vertical vibrations of sleepers generated by passing trains at various speeds. A model combining the track structure and the energy harvesting system was used. Results indicated the generated power was around 100 mW, assuming a 2 mm rail displacement amplitude at a frequency of 6 Hz. The presented track model was validated with UK network experimental data. Testing of a piezoelectric vibration cantilever harvester in publication [24] was focused on energy harvesting at a frequency of 5 to 7 Hz. An output power of 4.9 mW and a peak-to-peak voltage of 22.1 V were achieved on a rail vibrating with amplitudes of 0.2 to 0.4 mm at a frequency of 7 Hz. A design of a resonant electromagnetic harvester was published in papers [13,25]. An approach based on magnetic levitation was capable of energy harvesting at a broadband low-frequency vibration in range of 3 to 7 Hz. This device induced a peak-to-peak voltage 2.32 V and an output power of 119 mW when subjected to vibrations with 1.2 mm amplitudes and with an optimal resistive load of 44.6 Ohm.

An innovative approach was presented in paper [26], where authors deployed piezoelectric energy harvesting devices for monitoring a full-scale bridge structure undergoing forced dynamic testing by passing trains. A similar approach was used in paper [27], where the damage detection and structural health monitoring of a laboratory-scaled bridge was observed using a vibration energy harvesting device, in particular a cantilever-based piezoelectric energy harvesting device. The published approach had an advantage over the conventional accelerometer-based method in terms of power requirements, because energy storage and data transmission units were the only power-consuming parts of the system.

## 3. Model Based Design of Electromagnetic Trackside Energy Harvester

A passing train provides mechanical vibrations in rails and sleepers. The vertical deflection of a sleeper is depicted in Figure 1 using the variable *z*. This sag in a sleeper depends on the passing train’s mass, velocity and the quality of the rail subgrade. The proposed energy harvesting system is based on a principle of kinetic energy harvesting which can convert the kinetic energy from sleeper oscillations into useful electricity. A mechanical resonator is used for the transfer of input kinetic energy into the free oscillation of a seismic mass. A design of this kinetic energy harvester is based on a mass *m* which is suspended on two steel cantilevers with a known stiffness *k*_1_ and known mechanical damping *d_m_*. This longitudinal design could be placed on the top of a sleeper, or it could be embedded inside a new generation of innovative sleepers. 

On the basis of the previously published analysis, the electromagnetic energy transducer provides an effective harvesting power for this application. The oscillating mass is a part of a magnetic circuit, and its free oscillation *x* against a fixed coil generates useful electricity, which provides electromagnetic damping forces in the form of electrical damping *d_e_*.

### 3.1. Mathematical Model of One Degree of Freedom System

The kinetic electromagnetic energy harvesting system could be described by a multidomain model in the form of coupled mechanical and electrical systems, as depicted in Figure 2. The mechanical resonator is excited by ambient mechanical shocks *z* to the oscillating sleeper and this results in the relative movement *x*. The relative movement *x* of the mass *m* in the magnetic circuit consisting of the frame and a fixed coil is inversely proportional to the mechanical damping *d_m_*. Due to Faraday’s law, the relative movement of the magnetic circuit results in a change in the magnetic field of the coil, inducing an electromotive voltage *u_i_*. A model of an electromagnetic coupling coefficient was used for the description of the interaction between both the mechanical and electrical domains. The induced voltage depends on the design of the electromagnetic transducer (the electromagnetic coupling coefficient *c_EH_*) and its relative velocity. When a resistive electrical load *R_L_* is connected to a coil, then a current flows through the coil and electrical power is extracted from the system. The electrical power extracted from this system provides electromechanical feedback in a form of an electrical damping, which is depicted as a damper *d_e_*. This electrical damping feedback is proportional to the electromagnetic coupling coefficient *c_EH_*. A derived mathematical model with one degree of freedom was used for predicting the harvested power in a resonance operation.

A second-order equation according to the mechanical model in Figure 2 describes mechanical oscillations of a seismic mass as a response to the kinetic excitation of the sleeper:(1)$$m\ddot{x}+d_{m}\dot{x}+d_{e}\dot{x}+kx=-m\ddot{z}$$
where x is the relative displacement of the oscillating mass, z is the absolute displacement of the vibrating sleeper, m is the moving mass, $d_{m}$ is the mechanical damping, $d_{e}$ is the electrical damping, and k is the mechanical stiffness. 

The mechanical stiffness combines stiffness of both cantilevers. The stiffness of a single beam $k_{1}$ can be calculated using this equation:(2)$$k_{1}=\frac{3\cdot E\cdot J}{l^{3}}$$
where E is Young’s modulus of the used material (steel), J is a second moment of the area, and l is the length of the cantilever. The mechanical stiffness *k* is then simply $2\cdot k_{1}$ for this double suspended system.

The mechanical damping $d_{m}$ can be calculated using this relation:(3)$$d_{m}=\frac{1}{2Q_{m}}2m\Omega$$
where $Q_{m}$ is the mechanical quality factor, either estimated or calculated from an experiment. The natural frequency Ω can be calculated using a commonly known formula for a single degree of freedom system:(4)$$\Omega =\sqrt{\frac{k}{m}}$$

The electrical damping $d_{m}$ of the electromechanical system can be calculated using this equation:(5)$$d_{e}=\frac{{\left(BNl\right)}^{2}}{R_{C}+R_{L}}=\frac{{\left(c_{EH}\right)}^{2}}{R_{C}+R_{L}}\;$$
where B is the magnetic flux density in the coil, N is the number of turns, l is the active length of one turn, $R_{C}$ is the coil resistance, $R_{L}$ is the load resistance, and $c_{EH}$ is the electromagnetic coupling coefficient of the energy harvester, where $c_{EH}=BNl$.

The induced voltage on the coil $u_{i}$, can be calculated using equation:(6)$$u_{i}=BNl\cdot \dot{x}=c_{EH}\cdot \dot{x}$$

The first-order electric differential equation of the electrical circuit in Figure 2 is then:(7)$$L\cdot \frac{di}{dt}+i\cdot \left(R_{C}+R_{L}\right)=u_{i}$$
where L is the inductance of the coil, and i is the electric current. By design, the inductance of the coil in our harvester is very small (for a coil with an air core) and the current change is very slow, therefore the first term is irrelevant and the equation can simplified:(8)$$i=\frac{c_{EH}\cdot \dot{x}}{R_{C}+R_{L}}$$

The coupled mechanical equation can modified, where Equations (5) and (8) provide the electrical damping as a function of the electric current:(9)$$m\ddot{x}+d_{m}\dot{x}+c_{EH}i+kx=-m\ddot{z}$$

The fundamental performance of the energy harvester is the equation for the output power:(10)$$p_{out}=i^{2}\cdot R_{L}\;$$

The displacement amplitude (peak values) could be simply calculated from these equations assuming a resonance operation. The mechanical amplitudes of both the displacement and velocity follow these relations:(11)$$x_{A}=z_{A}Q_{T}=\frac{{\ddot{z}}_{A}}{\Omega^{2}}\;Q_{T}\to {\dot{x}}_{A}=\frac{{\ddot{z}}_{A}}{\Omega }\;Q_{T}$$
where the term $Q_{T}$ is the total quality factor of both the mechanical and electrical damping. This quality factor is a compound on the basis of the following relation:(12)$$Q_{T}=\frac{1}{2\frac{d_{m}+d_{e}\;}{2m\Omega }}=\frac{m\Omega }{d_{m}+d_{e}\;}$$

The calculation of the velocity amplitude can be used for the calculation of the amplitude of the induced voltage:(13)$$u_{i_{A}}=c_{EH}\cdot {\dot{x}}_{A}$$
and the amplitude of the output voltage on the resistive load is:(14)$$u_{L_{A}}=u_{i_{A}}\frac{R_{L}}{R_{C}+R_{L}}\;$$

The output power amplitude can then be expressed using either voltage or current:(15)$$p_{out_{A}}=i_{A}^{2}\cdot R_{L}=\frac{u_{L_{A}}^{2}}{R}\;$$

### 3.2. Design of Energy Harvesting Device

The proposed design of the electromagnetic kinetic energy harvester could be capable of converting sleeper vibrations into useful electricity. Resonance operation is not possible due to the pulse excitation characteristics produced by the passing train. However, the free oscillation response to the passing train provides a relative oscillation of the suspended seismic mass against the fixed base with a coil, which has the potential to generate satisfactory levels of useful electrical power. In the case of the longitudinal design of the device mounted on top of the sleeper, the suspension system consists of a pair of steel cantilevers with dimensions of 400 × 30 × 3 mm^3^. The mechanical resonator is by design tuned up to have a natural frequency of 12 Hz, which provides a sufficient relative movement. The long steel cantilever design results in a mechanical resonator with one degree of freedom in the vertical direction, which makes it sensitive to the train induced vibrations. The relative movement amplitude is important for a correct design of the magnetic circuit, which is fixed inside the seismic mass. A concept of a sleeper kinetic energy harvester design for trackside application is shown in Figure 3. 

The fundamental part of the seismic mass is a magnetic circuit with 16 rare earth FeNdB magnets and ferromagnetic holders. A pair of ferromagnetic holders with permanent magnets moves freely in the air gap of a fixed coil. A planar finite element analysis of this magnetic circuit was conducted in order to calculate the average magnetic flux density in the area of the coil for a given relative movement. The coil was designed to have an air core and wound up around a plastic frame fixed to the base. All active turns of the coil were placed in the air gap of the magnetic circuit. A relative position of the magnetic circuit and coil was set with a minimal air gap to achieve efficient electro-mechanical energy conversion. This analysis was done in an FEMM environment and the calculated magnetic field is shown in Figure 4.

The developed and assembled kinetic electromagnetic device for trackside application is shown in Figure 5 and it consists of:A base (1) fixed on a vibrating structure,The flexible suspension of a resonator (2)—its stiffness is provided by a pair of steel cantilevers,A resonator mass (3) with a magnetic circuit inside,A self-bonded air coil with a plastic coil holder (4).

The model from the previous chapter was used for the design of the individual parameters with respect to the required harvested power. The parameters of the final model and the assembled device (see Figure 5) are summarized in Table 1.

## 4. Electromagnetic Kinetic Energy Harvester Testing with Resistive Load

### 4.1. Resonance Operation: Model Results and Experiment

The designed parameters of the model are used to predict the output voltage and power in a resonance operation. The presented electro-mechanical equations in combination with the model of peak voltage and peak power described in Section 3.1 were used for output calculations for a variable resistive load. The experiment was conducted on a laboratory shaker, an RMS SW8142–SWH600APP, connected to its auxiliary measurement instruments and depicted in Figure 6. The harvested voltage was measured on an oscilloscope, a Rigol MSO 5204. Both the model and experiment were excited in a resonance frequency with an acceleration amplitude of 1 ms^−2^.

The calculated output voltage and power are shown in Figure 7, and both outputs are compared with values obtained from the experiment. The correlation between the model and experiment is very good for low values of the resistive load. Based on the harvester model, the maximal power was expected with a resistive load of 3 kΩ. However the experimental results showed that the maximal power was harvested for a resistive load of 2 kΩ, but the harvested power was very similar across a wide range of resistive loads, 2–3 kΩ. The experimentally measured voltage and power for a higher resistance were lower than the theoretical values and it seemed that the real damping was higher compared to the damping model at higher speeds, causing a less pronounced increase in the output voltage due to the voltage being proportional to the speed.

### 4.2. Test of Kinetic Energy Harvester—Harmonic Vibration

The frequency response is very important for characterizing an energy harvester’s performance. A shaker test with sinusoidal vibrations was used to measure the frequency response around resonance. Both sweep up and sweep down tests with a slowly changing input frequency of vibration were used for voltage and output power response measurements in the frequency domain. The input acceleration with an amplitude of 1 ms^−2^ and frequency rate of change of 0.1 Hz/s were used in the tests. These sweep tests were realized around the resonance frequency with different load resistance.

The measured peak voltage and peak power are depicted in Figure 8. The lab experiment was made with four different load resistances: 1, 2, 3 and 5 kΩ. The voltage response was proportional to the load resistance. It was caused by an increase in the velocity due to decreasing electrical damping. However, the output power peaks at around 2 kΩ, and below and above that value the power amplitudes decreased. This power maximum was observed in a resonance operation, but outside of the resonance operation the device harvested higher power with the lowest load resistance of 1 kΩ. The kinetic energy harvester with a lower resistance provided higher power outside of a resonance operation. Based on this fact, it is suitable to use a resistive load of 2 kΩ in a resonance operation but use a lower value of 1 kΩ for a non-resonance operation. This fact is discussed further in the next section.

### 4.3. Test of Kinetic Energy Harvester—Train-Induced Vibrations

The typical transient dynamic response of a kinetic energy harvester was provided with real train-induced vibrations as inputs. Real vibrations in the trackside sleeper, where the vibrations were not pure sinusoidal, but rather had the characteristic of a series of mechanical pulses, are shown in Figure 9. The acceleration and displacement of the sleeper was measured on a trackside in the Czech Republic using an inertial accelerometer on the sleeper and a capacitive displacement sensor mounted between the sleeper and a fixed point. This detail is for a single bogie consisting of two train wheels. The movement of the sleeper had general characteristics based on many parameters of the track subgrade. For this particular sleeper movement, the shown input vibrations for whole trains are used for the kinetic energy harvester test. Experiments with different electric loads were made with the aim of finding an optimal resistive load for a maximal energy harvesting potential.

Real acceleration and displacement measurements of sleeper movement for two typical trains were used for lab tests of the developed kinetic energy harvesters:Train 1—Regional train travelling at 80 km/h,Train 2—Express train travelling at 130 km/h.

As Figure 10 shows, the maximal average power was measured with a resistive load of 150 Ω, which was the same as the coil resistance. The output power was lower at higher resistive loads due to lower damping and lower energy harvesting from trackside vibrations. The power was lower at lower resistive loads due to higher energy dissipation on the coil resistance rather than the load resistance.

Different weights of the seismic mass of this device were tested in order to achieve the optimal design of this energy harvesting device. The default weight of the moving mass, 800 g, was increased using an additional external weight. Unfortunately, decreasing the weight below the default value without significant structural modifications was not possible. A relation of this mass value to the generated power was experimentally verified, and the results of his test are shown in Figure 11. It was possible to add an extra mass to the initially calculated seismic mass of 800 g; however, it is evident that the output power would decrease with higher values of the seismic mass, and it was verified that the modelled mass of 800 g provided the most effective energy harvesting operation with the given train vibration data. On the other hand, it was not possible to decrease the mass in the current design due to the nature of the magnetic circuit, which needs a minimal cross-section area of the core to function properly.

Time-domain measurements of the voltage and power with the excitation acceleration and displacement for both testing trains are shown in Figure 12. These measurements were done with the optimal resistive load of 150 Ω. The average harvested power and total harvested energy for a passing train are presented in Table 2.

Both the voltage and power strongly depended on the input acceleration during the train’s passage. While the average power was 9.1 mW for the regional Train 1, the maximum power during much more pronounced acceleration peaks was up to 300 mW. In the case of the express Train 2, this vibration could generate average power of 29.3 mW and several peaks above 600 mW could be observed. Given all the information it is necessary to mention that the variable nature of the output power must be considered during the design of power management electronics for railway applications.

## 5. Kinetic Energy Harvester as Source of Energy for Wireless Sensing

The previous chapter shows that the kinetic energy harvester could generate useful electric power from passing trains. However, a test with power management electronics, a sensing node, and a communication module is required to fully assess the feasibility of this energy harvesting technology. For this reason, operational tests of the kinetic energy harvester utilized as a source of energy for a wireless sensor node were conducted for both types of trains. Our lab successfully tested autonomous wireless sensor nodes with the vibration energy harvester in lab conditions, and these results, including the design of a power management, sensing unit and communication module, were successfully published in our research paper [28]. 

The electric diagram of an energy harvesting system with a sensor node and communication module is shown in Figure 13. The developed kinetic energy harvester is connected to previously published electronic systems. It consists of a rectifier, a storage capacitor with a capacity of 493 μF, an LTC 3588 power management circuit, an analogue front end for the sensing signal, and a communication module based on the NRF24L01 chip from Nordic Semiconductor. 

Measured vibration signals for both types of passing trains (the same as those in the previous section) were used for the test of this autonomous source of energy for the sensing node and communication module. Measurements of electrical signals highlighted in Figure 13 (the voltage induced by the energy harvester, the input voltage and the current into the LTC power management circuit of the wireless sensor node) were used for the harvester performance assessment.

Experimental results of the autonomous wireless sensing node operation are depicted in Figure 14 for both train types. The voltage induced on the developed kinetic energy harvester was very similar to the response with a resistive load. The input voltage into the LTC circuit rose quickly during the first second of the train’s passage and then began to fluctuate around a constant value for the regional Train 1, or slowly increase for the express Train 2, which provided more power in general. After a train passed, the voltage decreased slowly and transmission continued. The current was discontinuous due to characteristics of the LTC power electronics, and this pattern reflected the transmission of the measured data.

During this experiment, the average power consumption of the radio transmission unit was around 4.5 mW, which was less than the average power acquired from each train. The power harvested during a train’s passage was stored in the capacitor, and after the passage it was used to continue the data transmission. This energy could also be stored for later if the transmission was no longer active after the respective train passed. However, it would strongly depend on the sensor setup and monitoring requirements for any given application. Nevertheless, experiments with both types of trains confirmed that the proposed system was able to provide enough power for continuous transmission, as is evident in Figure 14.

Measurement with different capacitor values are depicted in Figure 15. The excitation acceleration data were acquired from Train 1, similarly to Figure 14a. For a better understanding of the energy storage capabilities, the graph of the voltage is accompanied by a graph depicting the calculated energy stored in the capacitor. Initially, the lower capacity caused the voltage to increase rapidly; however, after a few seconds of operation the stored energy was lower compared to the case with higher capacity. Furthermore, with lower capacity values the voltage tended to fluctuate significantly.

## 6. Potential Applications

Several manufacturers of railway infrastructure systems (e.g., sleepers/bearers, railway switches, point machines, axle counters etc.) have begun to seriously consider the development of new equipment with energy harvesters for predictive maintenance applications of their products. There are two potential workstreams: retrofitting existing products, and novel design of products with embedded energy harvesting. Both emphasize the main advantages of energy harvesting devices, which would result in a reduction in wiring and cables (in both communication and power), reduction of losses due to cable theft, decreased costs of the energy power supply and battery replacement etc.—and all of these aspects contribute to a significant reduction in maintenance costs. This fact is amplified by the subsequent application of predictive maintenance methods, which replace periodic inspections and, compared to the conventional methods, can reveal potential defects and abnormal degradation before a fatal system failure occurs.

### 6.1. Smart Railway Monitoring Application

A new generation of smart sleepers could detect overloaded trains, trains with abnormal wheel wear or problems with suspension systems, all of which contribute to a significantly higher load on the track and a rapidly increasing wear of railways. The railway wear or significant changes in subgrade and ballast properties—mainly the gap under a sleeper—could also be detected by smart monitoring systems embedded in sleepers. The proposed concept of autonomous application is shown in Figure 16. The kinetic energy harvester could be integrated or embedded inside innovative sleepers and provide electricity for autonomous monitoring. The sensing node could be integrated into rails in the form of a piezoelectric layer that generates an active voltage signal and does not consume power. This system could be even more affordable if PVDF piezopolymers for structural monitoring were used.

This maintenance system could be interesting for infrastructure management and freight train providers interested in detecting critical wear or damage of both the railway and trains. The comprehensive monitoring system could solve the sensitive question of whether freight cars are subjected to excessive wear due to poor track quality, or conversely whether damaged freight cars in operation are causing excessive wear of railways. 

### 6.2. Concept of Smart Turnouts

Switches and crossings are the parts of a railway track most impacted by the dynamic forces applied by trains. From a maintenance point of view, it is important to recognize faults or degradation processes at an early stage, before any significant limitation in operability occurs. The best way to monitor the conditions of crossings is with condition monitoring systems capable of measuring and evaluating the dynamic impacts on crossings over longer periods of time. The concept of an autonomous sensing node with a vibration energy harvester could represent a suitable solution for this system, and kinetic energy harvesters could provide sufficient energy, especially if piezopolymer materials were utilized as the active piezoelectric sensors (e.g., PVDF), mainly because they do not require an external power source to operate. The concept of an autonomous wireless node depicted in Figure 17 can transmit signals from the turnout structure to a close trackside IoT point. There is also the option to process signals on-site and transmit only the results of embedded data analyses. Energy harvesting could mostly be useful in the case of short-distance wireless communication between the track structure and the IoT point, which would use electricity from the power grid.

## 7. Conclusions

The main design goal of all energy harvesting devices should be to deliver sufficient power for the operation of a given system. In the case of railway applications, replacing cables, cutting off railway systems from the power grid, and using energy harvesting sources for every system is not always an optimal solution, and in many cases is nearly impossible to implement. It is important to keep in mind that energy harvesting devices are utilized as power sources for specific tasks and must be developed and designed with respect to the system they are used to power in order to achieve reliable, low-power and maintenance-free operation over long periods of time. Only such an approach would inevitably lead to a long-term deployment of energy harvesting devices with a new generation of smart railway systems and parts for sustainable rail transportation.

The maintenance of cable systems could result in damage to the wires used in wired sensing systems. For this reason, using embedded kinetic energy harvesters as a source of energy for autonomous wireless sensing nodes is an advantageous approach for long-term railway sensing and monitoring. Two potential applications are presented in this paper for smart rail monitoring and a turnout predictive maintenance system.

The main aim of this paper was to present the development of a kinetic energy harvesting device for rail track applications, a device that is able to provide sufficient power for short-distance communication. The developed device was tested under lab conditions with the input vibration signals of two different trains, regional and express. In lab tests, vibrations obtained from a real rail track used as an input provided enough energy for the communication module and transmission of the sensing signal, with the results presented in this paper. The concept it uses, of placing a kinetic energy harvester on the top of sleepers, could generate an average output power in range of 5–35 mW, depending on the train speed. In this case, this technology could be attractive for the retrofitting of the existing railway infrastructure and for innovative products.

## Figures and Tables

**Figure 1 sensors-22-00905-f001:**
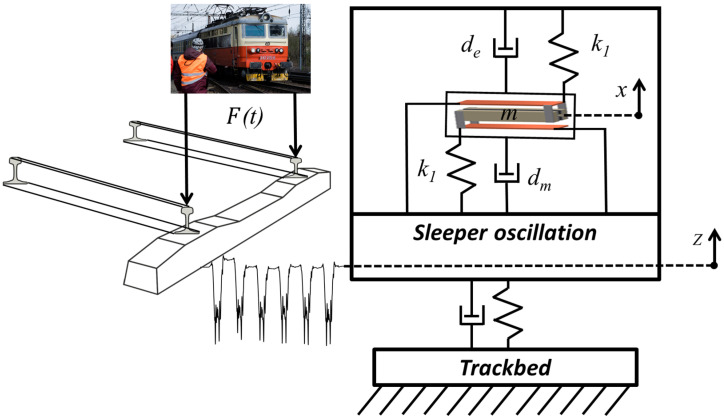
Physical principle of kinetic energy harvester under passing train vibrations.

**Figure 2 sensors-22-00905-f002:**
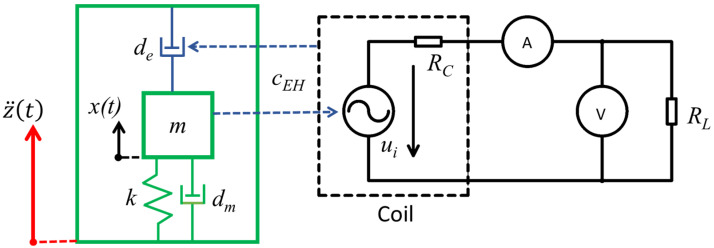
Coupled mechanical and electromagnetic models of the proposed kinetic energy harvester.

**Figure 3 sensors-22-00905-f003:**
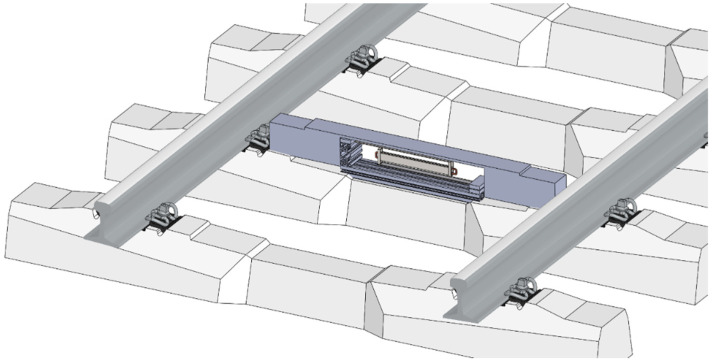
Proposed integration of KEH design for railway applications.

**Figure 4 sensors-22-00905-f004:**
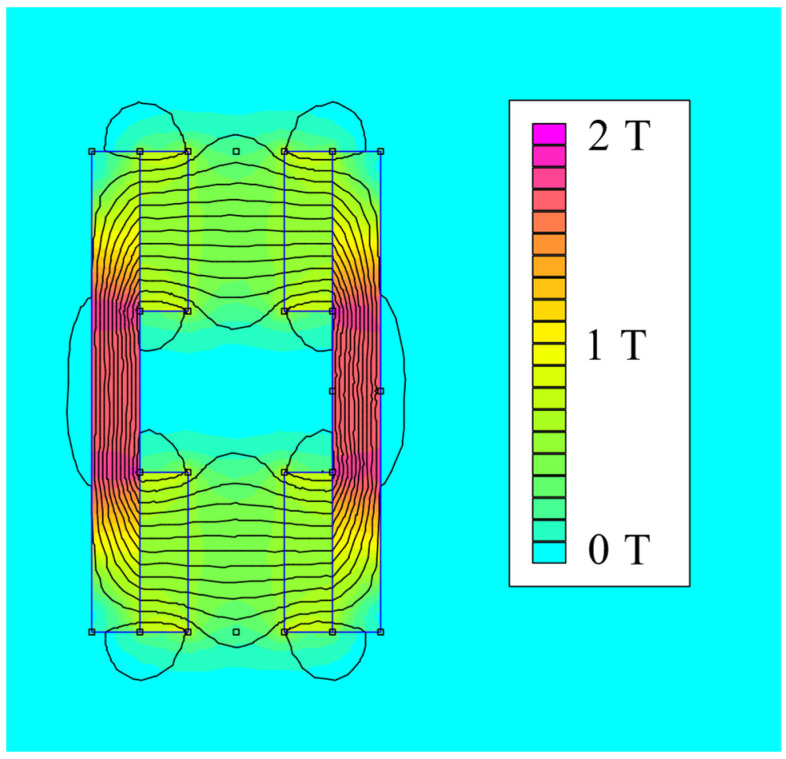
Planar FEMM model of magnetic circuit; analysis of magnetic flux density *B*.

**Figure 5 sensors-22-00905-f005:**
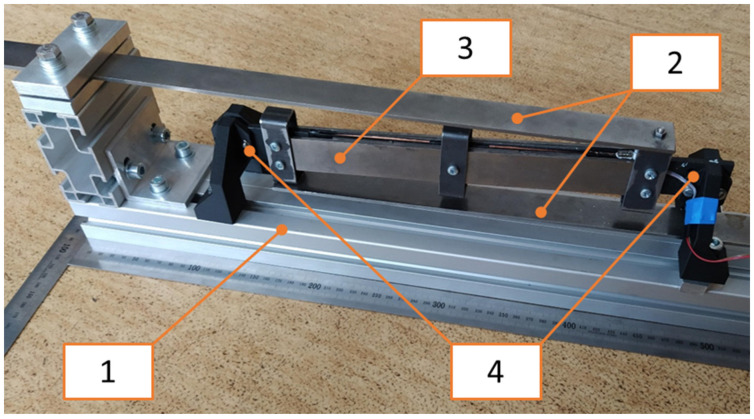
Design of proposed kinetic energy harvester.

**Figure 6 sensors-22-00905-f006:**
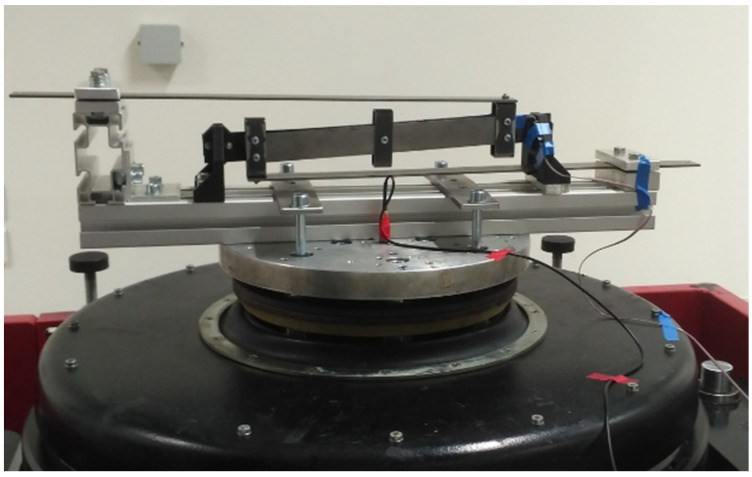
Shaker lab test of kinetic energy harvester.

**Figure 7 sensors-22-00905-f007:**
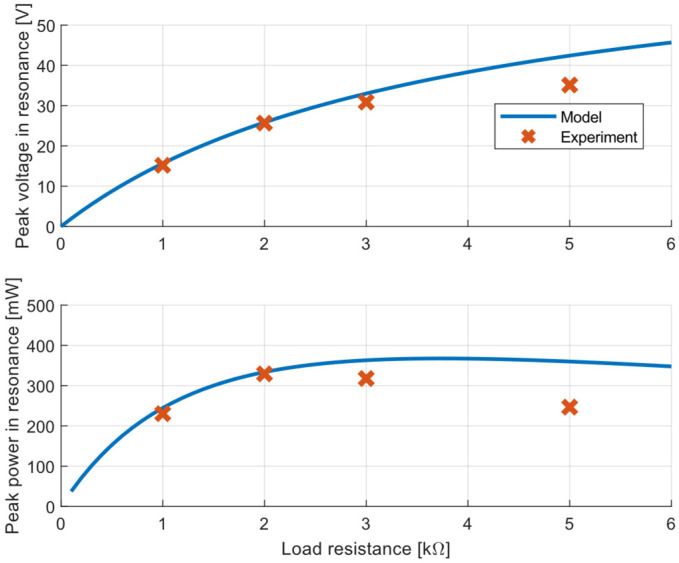
Voltage and power responses in resonance operation vs. load resistance—simulation results and measurements with excitation acceleration amplitude of 1 ms^−2^.

**Figure 8 sensors-22-00905-f008:**
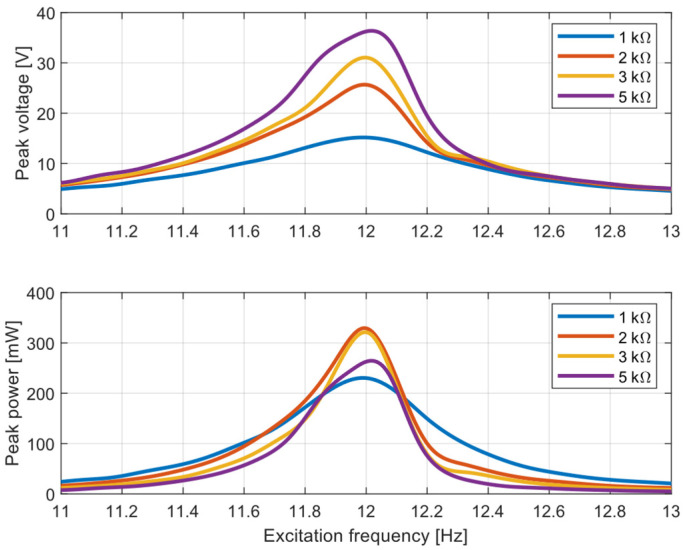
Voltage and output power depending on excitation frequency with different load resistance. Appendix A.

**Figure 9 sensors-22-00905-f009:**
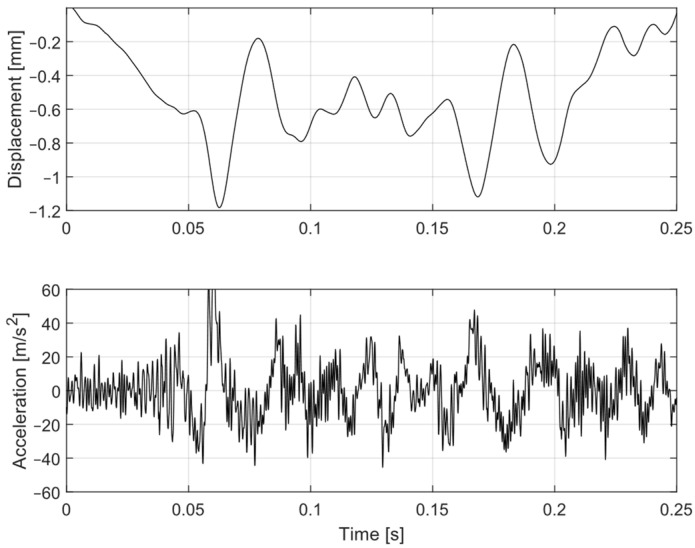
Typical vibrations of sleeper under passing train—detail of bogie/two wheels.

**Figure 10 sensors-22-00905-f010:**
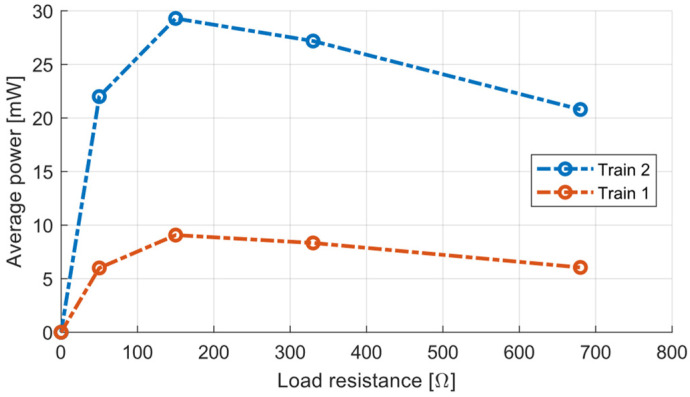
Average power depending on load resistance for Train 1 and Train 2.

**Figure 11 sensors-22-00905-f011:**
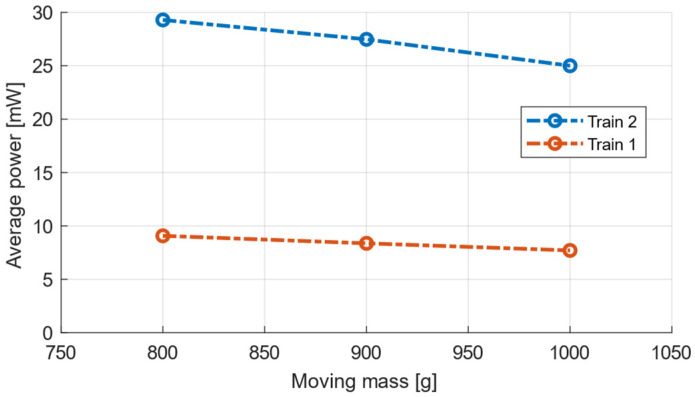
Average power depending on moving mass for Train 1 and Train 2.

**Figure 12 sensors-22-00905-f012:**
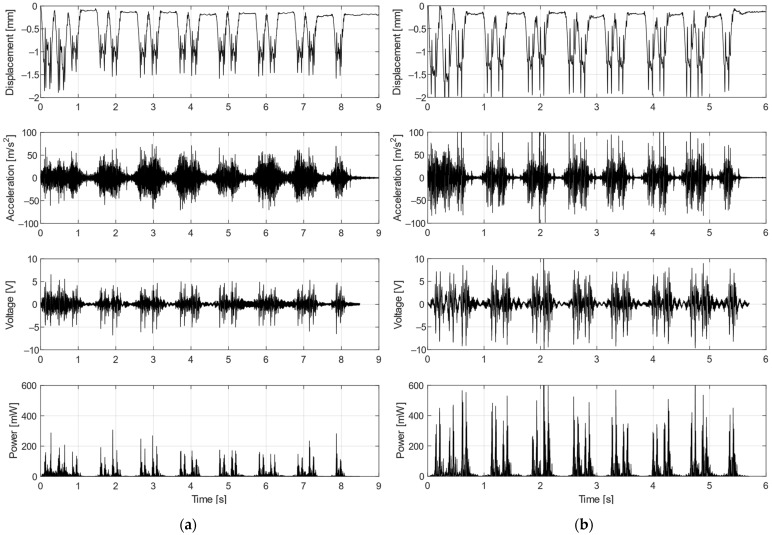
Lab shaker test with real acceleration data from regional Train 1 (**a**) and from express Train 2 (**b**); measured voltage and power with load resistance of 150 Ω for input mechanical vibrations of the shaker.

**Figure 13 sensors-22-00905-f013:**
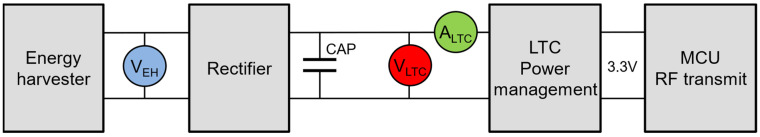
Schematic diagram of energy harvesting system with highlighted electrical parameters measurement.

**Figure 14 sensors-22-00905-f014:**
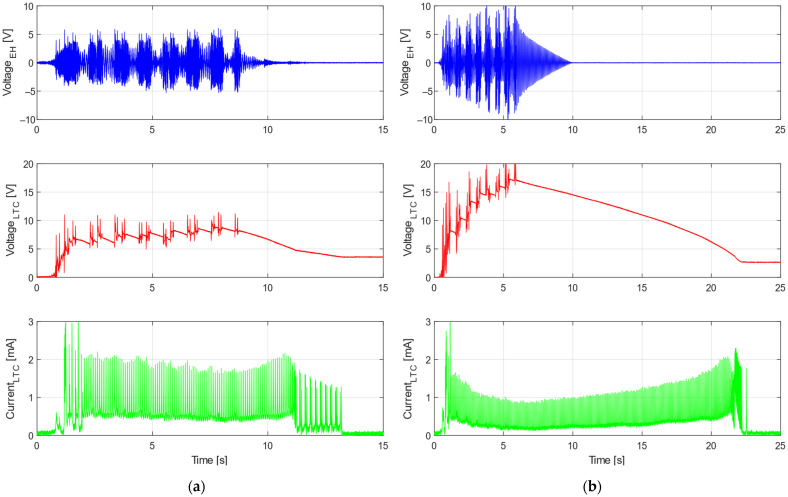
Shaker test with real acceleration data from Train 1 (**a**) and Train 2 (**b**) with connected power electronics; measured voltage and current according to electric schema. Appendix A.

**Figure 15 sensors-22-00905-f015:**
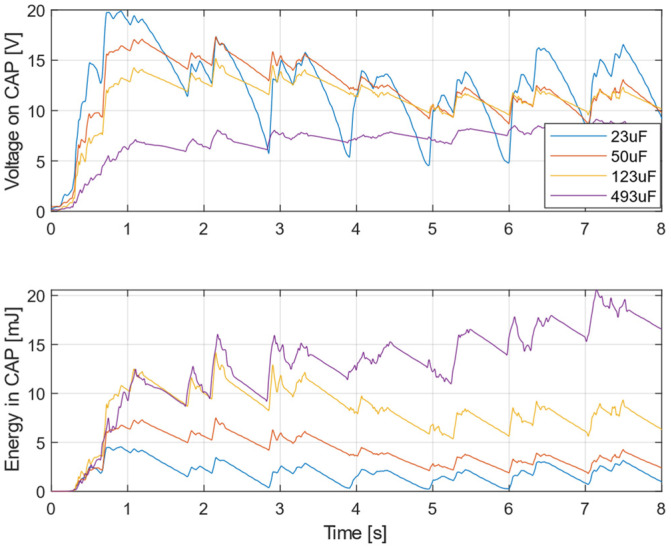
Charging process of capacitor of different value with continuous power consumption of wireless sensor.

**Figure 16 sensors-22-00905-f016:**
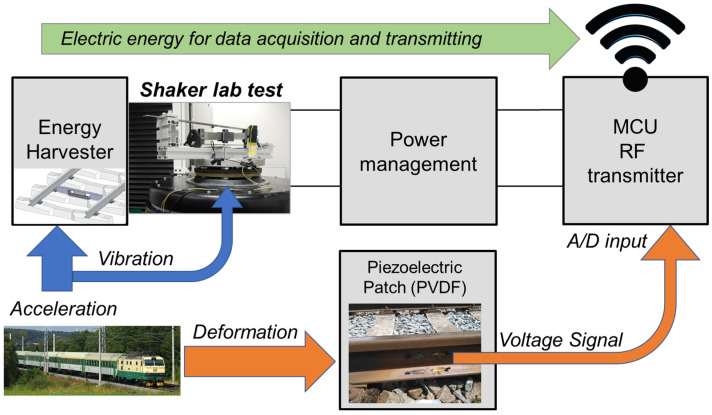
Proposed autonomous railway application of autonomous piezoelectric sensing. Appendix A.

**Figure 17 sensors-22-00905-f017:**
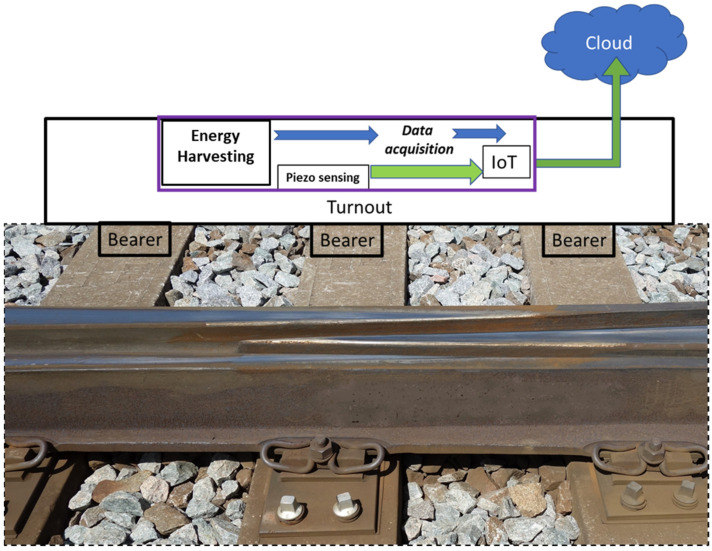
Proposed concept of smart turnout based on energy harvesting.

**Table 1 sensors-22-00905-t001:** Parameters of individual harvesters used in experiments.

Parameter	Symbol	Value
Total weight	-	3.9 kg
Total dimensions	-	600 × 160 × 90 mm^3^
Moving mass	m	0.8 kg
Resonance frequency	Ω	12 Hz
Mechanical quality factor	$Q_{M}$	150
Coil dimensions	-	210 × 25 × 3 mm^3^
Coil wire diameter	-	0.15 mm
Coil turns	N	300
Coil resistance	$R_{C}$	150 Ω
FeNdB magnetic circuit dimensions	-	Two pairs, 3 × 10 × 180 mm^3^
Air gap	-	6 mm
Average magnetic flux density	B	0.3 T

**Table 2 sensors-22-00905-t002:** Harvested power and energy with optimal load resistance.

Acceleration Source	Average Power	Harvested Energy
Regional Train 1	9.1 mW	77 mJ
Express Train 2	29.3 mW	167 mJ

## Data Availability

Not applicable.

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
