# Peer review of "Kinetic Electromagnetic Energy Harvester for Railway Applications—Development and Test with Wireless Sensor"

_sensors, 2022, doi:10.3390/s22030905_

Round 1

Reviewer 1 Report

This manuscript reports the development and lab testing of energy harvesting technology for autonomous sensing in railway applications.

The manuscript is consistent, well organized, very well written and present a significant contribution to the field of wasted energy harvesting.

Can the authors provide some video to better illustrate the experimental process?

For practical application, it is required to include some usual energy harvesting tests like lightning LEDs and charging capacitors for example.

Author Response

General comment: This manuscript reports the development and lab testing of energy harvesting technology for autonomous sensing in railway applications.

The manuscript is consistent, well organized, very well written and present a significant contribution to the field of wasted energy harvesting.

General response: Thank you for the comments.

Point 1: Can the authors provide some video to better illustrate the experimental process?

Response 1: It is very good point; videos were included for both resonance and train operation.

Point 2: For practical application, it is required to include some usual energy harvesting tests like lightning LEDs and charging capacitors for example.

Response 2: Thank you for your comment. The test with different capacitors was added in Figure 15 and lines 413-422.

Reviewer 2 Report

Review Comments

In the manuscript entitled “Kinetic Electromagnetic Energy Harvester for Railway Applications – Development and Test with Wireless Sensor,” the authors designed and tested a kinetic electromagnetic energy harvesting system for harvesting energy from a passing train. The harvested energy is used to power sensing nodes. However, after a thorough review following are some comments and suggestions to improve the manuscript.

  1. The author should provide the instrument details used for all the experiments in the manuscript.
  2. The author should clearly mention how the measurement for figure 9 was carried out.
  3. The experiment using various resistive load, more load resistance should be tested in the given range
  4. For figure 11, mass lower than 800 g should also be tested to improve the quality of the manuscript.
  5. In section 5, the harvester was connected to a previously published electronic system. The video proof for the system's working using the reported harvester should be included.
  6. The author should give the reason for using an acceleration amplitude of 100g for the testing while the rest of the experiments were at lower acceleration as per the data of the real train.
  7. Carefully go through the entire manuscript for spelling mistakes. These are a few, line 123- “Abovementioned” → Above mentioned, Line 338- “the results of his test” → the results of this test.

Author Response

Please see enclosed document with response.

Round 2

Reviewer 1 Report

All the changes requested by the reviewer have been introduced by the authors. Therefore, I propose the acceptance of the manuscript in its current form.

Author Response

Thank you very much for your effort.

Reviewer 2 Report

Review Comments

In the manuscript entitled “Kinetic Electromagnetic Energy Harvester for Railway Applications – Development and Test with Wireless Sensor,” the authors designed and tested a kinetic electromagnetic energy harvesting system for harvesting energy from a passing train. The harvested energy is used to power sensing nodes. During the revision, the authors have been asked to provide technical details regarding the experiments, video proof for the applications claimed, explanation for the use of specific parameters, and make proper corrections throughout the manuscript. However, the authors failed to explain some of the review comments as listed below.

  1. More load resistance should be tested to understand the simulation and the experiment result better.
  2. The authors claimed to have powered an electronic device using the harvester in the manuscript. Video proof was requested, but the authors provided a video of the harvester alone. Video proof for the harvester powering the electronic device should be provided.

Author Response

Please check document with our response.
